# An Empirical Investigation of “Physician Congestion” in U.S. University Hospitals

**DOI:** 10.3390/ijerph16050761

**Published:** 2019-03-02

**Authors:** Eran Manes, Anat Tchetchik, Yosef Tobol, Ronen Durst, Gabriel Chodick

**Affiliations:** 1The Department of Public Policy and Administration, Ben-Gurion University of the Negev, P.O. Box 653, Beer-Sheva 84105, Israel; msemanes@gmail.com; 2Faculty of Management, Lev College of Technology, Havaad Haleumi 21 St., Givat Mordechai, Jerusalem 9116001, Israel; tobol@jct.ac.il; 3The Department of Geography and Environment, Bar-Ilan University, Ramat-Gan 5290002, Israel; 4IZA—Institute of Labor Economics Schaumburg-Lippe-Straße 5-9, 53113 Bonn, Germany; 5Cardiology Division, Hadassah Hebrew University Medical Center, Ein Kerem, Jerusalem 91120, Israel; Durst@hadassah.org.il; 6School of Public Health, Sackler Faculty of Medicine, Tel Aviv University, P.O. Box 39040, Tel Aviv 6997801, Israel; hodik_g@mac.org.il

**Keywords:** health care quality, physicians per bed, doctors per bed, clinical performance, increasing returns, inverted U-shape

## Abstract

We add a new angle to the debate on whether greater healthcare spending is associated with better outcomes, by focusing on the link between the size of the physician workforce at the ward level and healthcare results. Drawing on standard organization theories, we proposed that due to organizational limitations, the relationship between physician workforce size and medical performance is hump-shaped. Using a sample of 150 U.S. university departments across three specialties that record measures of clinical scores, as well as a rich set of covariates, we found that the relationship was indeed hump-shaped. At the two extremes, departments with an insufficient (excessive) number of physicians may gain a substantial increase in healthcare quality by the addition (dismissal) of a single physician. The marginal elasticity of healthcare quality with respect to the number of physicians, although positive and significant, was much smaller than the marginal contribution of other factors. Moreover, research quality conducted at the ward level was shown to be an important moderator. Our results suggest that studying the relationship between the number of physicians per bed and the quality of healthcare at an aggregate level may lead to bias. Framing the problem at the ward-level may facilitate a better allocation of physicians.

## 1. Introduction

According to the U.S. Centers for Medicare & Medicaid Services, total healthcare spending for 2016 in the U.S. reached $3.3 trillion, up 4.3 percent from 2015. It is projected that spending on healthcare will surpass $5.5 trillion by 2025 [1]. Considering this ever-increasing expenditure, the relationship between healthcare spending and clinical performance is a topic of keen interest and is stimulating heated public discourse among proponents of more healthcare spending and supporters of “bending the cost curve”. The question remains as to whether more intensive spending is necessary to the delivery of high-quality healthcare, or whether greater efficiency gained from a reallocation of existing resources may lead to better results.

Most of the relevant empirical literature focuses on the link between different measures of healthcare spending (total spending, Medicare spending), usually at the cross-country, state or regional level, and alternative measures of healthcare quality [2,3,4,5,6,7]. A large body of work, led mostly by researchers of the Dartmouth Group, has provided consistent support for the view that more spending is either neutral or even counter-productive when it comes to achieving better healthcare results [1,2,3,4,5,6,7,8,9,10,11]. On the other hand, other studies support a positive correlation between spending and healthcare quality [12,13,14]. A recent study employed a meta-regression analysis of 65 studies and found small spending elasticities, implying that limits exist for the extent to which healthcare spending can improve health results [15]. For instance, it was predicted that while healthcare spending as a percentage of GDP has almost doubled among OECD countries since 1970, mortality fell by only 8% holding other factors equal.

In this paper we explore yet another angle of the link between spending and performance by focusing on the relationship between the (correctly adjusted) number of physicians at the hospital ward level and the resulting quality of healthcare, as measured by various clinical scores (to the best of our knowledge, the first study to measure the relationship between healthcare expenditure and performance at the ward level was conducted in 2011 [16]).

There are several reasons why exploring the link between physician workforce size and medical performance at the ward level is important:Increasing returns to scale resulting from specialization and internal knowledge spillovers are more likely to take place in small teams and units within organizations. In light of this, analysis of state-level spending on the overall physician workforce vis-à-vis aggregate measures of healthcare quality, while important in its own right, could omit one of the most important processes that links scale and size (of organizations) with performance and productivity.Many capital resources such as new laboratories or radiotherapy centers, which may have direct effects on healthcare quality, require substantial time to install (meaning that they are fixed inputs, at least in the short run). However, the number of affiliated physicians is a variable input that can be adjusted relatively easily in the short run.While state of the art medical technology such as Premium CT Scanners, proton beam therapy systems, and MRI scanners exerts a substantial positive net effect on healthcare, the enormous costs means that the services they provide are usually offered in a limited number of regional or national tertiary referral centers. This provides little room for efficiency gains through redeployment and reallocation. On the other hand, physicians can relocate with relative ease. Understanding the effects of physician relocation, thus, could create a better understanding of the nexus between efficiency and resource reallocation. There is a growing body of literature, related to our paper, that explores the relationship between physician volume and clinical outcomes at the hospital level. In this vein, other studies have examined the effects of physician volume on readmission and mortality in elderly patients with heart failure. They found that patients treated at hospitals with low physician volumes had higher readmission and mortality rates than those treated at hospitals with high physician volumes [17,18]. Controlling for patient, physician, and hospital characteristics they showed that patients with heart failure cared for by the high-volume physicians had lower mortality than those cared for by the low-volume physicians [18]. Focusing on elderly patients with pneumonia, a recent study found that patients cared for in hospitals with more doctors were less likely to be readmitted [19]. The joint effect of hospital and physician volume of primary percutaneous coronary intervention (PCI) on in-hospital mortality was also examined. This study found that primary PCI by high-volume hospitals and high-volume physicians was associated with lower mortality risk [20]. Finally, Birkmeyer et al. tested the link between surgeon volume and operative mortality in the United States for patients who underwent one of eight cardiovascular procedures or cancer resections. For all eight procedures, surgeon volume was found to be inversely related to operative mortality [21].

However, none of these papers, considered the possibility that the relationship between physician volume and clinical outcomes is non-linear. These studies might have overlooked the possible existence of non-linearities owing to limited scale of the (correctly adjusted) physician volume. Several theoretical arguments drawn from elementary theories in economics and organization science, R&D performance, and the biomedical academic literature led us to propose that the ward-level relationship between spending on medical workforce expansion and outcomes would be hump-shaped. These arguments are:

(1) Internal knowledge spillovers, division of labor and increasing returns to scale. A vast literature on economic growth and R&D performance indicates a strong tendency for geographical clustering in high-tech and knowledge-intensive sectors [11]. Internal knowledge spillovers, i.e., the tendency of knowledge, know-how and techniques to spill over from one team member to another, are a strong driver for increasing returns. Starting from the famous pin factory example of Adam Smith, it is widely agreed that increasing returns are also the result of specialization and the division of labor. The latter two processes tend to grow with organizational scale up to a point, simply because an expansion of the workforce allows better utilization of task division, and hence greater specialization.

(2) Organizational limits to increasing returns. While performance is expected to increase with scale, at least within some initial range due to scale economics, this increase cannot continue indefinitely. In his influential 1937 paper Coase, a Noble Prize laureate in Economics, answered the question of why production in the economy is not dominated by a single firm. He mentioned organizational costs such as policing and monitoring, and coordination costs as limits to organizational size [22]. Other studies reinforced this argument by stating that inefficiencies in the decision making of large organizations act as natural boundaries to further size increases [23,24]. In a context more pertinent to the choice of an optimal physician workforce, complexity, i.e., the probability of harm due to a failure in any single step of a common treatment—which arises when task division becomes complicated—is one of the potential disadvantages associated with an excessive expansion of the physician workforce [16]. One of the leading causes of medical errors that this literature stresses is a breakdown in communication, which is clearly associated with the expansion of the workforce.

Combining the arguments above, and in line with Coase’s hypothesis, we posited that:

**Hypothesis** **1:**
*The relationship between department size, defined as the number of physicians-per-bed, and the quality of healthcare is hump-shaped.*


As a testing ground, we used a cross-section of 150 U.S. university hospital departments across three different specialties—orthopedics, oncology and cardiology. The justification for choosing these specialties is stated in Section 2.1. The data offer a comparison between for-profit and nonprofit hospitals, while also adjusting for income, medical services and other important controls, making it economically viable to deal with efficiency issues. While running a comprehensive and properly controlled experiment is difficult in our context, the large set of covariates we controlled for, and the cross-sectional diversity in both the physicians-per-bed ratio and the quality of healthcare among hospitals in the sample enabled us to test the relationship properly and reliably.

Our focus on university hospitals allowed us to account for the biomedical research output conducted by physicians, a measure which was found to be an important explanatory variable for healthcare quality [25,26]. Thus, we could test whether the quality of bio-medical research conducted by the physicians at the ward level moderated the relationship between healthcare quality and the physicians-per-bed ratio (while research quality refers both to theoretical and applied research, it seems likely that in the context of physicians in hospital it is mainly applied research that directly contributes to clinical performance). To the extent that a synergy exists between bio-medical research quality and practice, we expected that:

**Hypothesis** **2:**
*Research quality strengthens the relationship between the number of physicians and the quality of healthcare.*


In other words, research quality was expected to raise the “marginal productivity” of physicians, and hence strengthen the effect of adding a single physician on the quality of healthcare.

As we report in the Results section, the empirical evidence supported both of our hypotheses.

Our paper contributes to this literature in several important ways: firstly, it is the first to hypothesize and empirically validate the existence of a hump-shaped relationship between physician (per bed) volume and quality of healthcare; and secondly, by employing proxies for biomedical research quality, we show that the latter is an important moderator for the relationship between physician volume and healthcare results.

Our approach, with the emphasis it places on the link between scale and performance, presents a shift in the empirical methodology relative to existing studies that have focused on the relationship between overall healthcare expenditure and outcomes. With respect to healthcare expenditure, most previous studies attempted to obtain measures of overall costs by using input and output prices, or aggregate data derived from sources such the Medical Expenditure Panel Survey. With respect to performance, many of these studies focus on nursing homes, for which a publicly available ranking of healthcare quality does not exist. Our methodological approach did not require the estimation of overall healthcare costs, simply because our variable of interest was the physicians-per-bed ratio at the ward level. Finally, our approach allowed us to use an independent measure of healthcare performance since our sample was made up of U.S. university hospitals for which a widely accepted ranking of healthcare quality was already available (see Materials and Methods section for more details).

## 2. Materials and Methods

### 2.1. The Empirical Model

Considering the discussion in the introduction, we hypothesized the following reduced-form relationship between the quality of healthcare results and a set of labor and capital outputs:(1)$$y_{ij}=\;A_{ij}f_{ij}\left(n_{ij}\right)g_{ij}\left(x_{ij}\right)$$

Here $y_{ij}$ is a scalar measure of healthcare quality in specialty i at hospital j, $A_{ij}$ captures a specialty-specific productivity factor, and $n_{ij}$ is the adjusted number of physicians. The function $g_{ij}$ is a Cobb-Douglas production function that depends on a vector $x_{ij}$ of capital inputs and other relevant factors known to affect the quality of healthcare. The multiplicative form of (1) captures the idea of complementarity between labor and other inputs.

Note that for the specifications in Equation (1) to be consistent with our hypothesis, the function $f_{ij}$ must attain a maximum at some interior point $n_{ij}=n_{ij}^{*}$. The reduced-form production function in (1) therefore captures our main hypothesis of a non-linear relationship between healthcare results and the adjusted workforce size.

To proceed, we log-linearize Equation (1):(2)$$ln\;\left(\;y_{ij}\right)=\;\ln(A_{ij})+lnf_{ij}\left(n_{ij}\right)+lng_{ij}\left(x_{ij}\right)$$

We further assume that $lnf_{ij}\left(n_{ij}\right)$ admits a Taylor approximation near $n_{ij}^{*}$, given by $lnf_{ij}\left(n_{ij}\right)=a_{0}+b{\left(n_{ij}-n_{ij}^{*}\right)}^{2}$ (higher order polynomials in $\left(n_{ij}-n_{ij}^{*}\right)$ are subsumed in the error term):(3)$$\;ln\;\left(\;y_{ij}\right)=\;\ln(A_{ij})+a_{0}+\;b{\left(n_{ij}-n_{ij}^{*}\right)}^{2}+lng_{ij}\left(x_{ij}\right)$$

Here, $a_{0}=\ln\left(f\left(n_{ij}^{*}\right)\right)$ and $b=\frac{f^{\prime \prime }\left(n_{ij}^{*}\right)f\left(n_{ij}^{*}\right)}{2!f{\left(n_{ij}^{*}\right)}^{2}}$ are constants. With $f_{ij}$ assumed to attain a maximum at $n_{ij}^{*}$, we must have $f^{\prime \prime }(n_{ij}^{*})<0$ so that b is expected to have a negative sign.

Using $a_{ij}\equiv \ln(A_{ij})+\ln\left(f\left(n_{ij}^{*}\right)\right)$ and drawing on the Cobb-Douglas from of $g_{ij}$ (so $g_{ij}=\prod_{k=1}^{N}x_{ijk}^{\alpha_{k}}$ and $lng_{ij}=\sum_{k=1}^{N}\alpha_{k}x_{ijk}$, and where $x_{ij}=\left(x_{ij1},x_{ij2},\ldots ,x_{ijN}\right)$ is a vector of inputs), the equation we want to estimate is therefore:(4)$$ln\;\left(\;y_{ij}\right)=\;a_{ij}+b{\left(n_{ij}-n_{ij}^{*}\right)}^{2}+\overset{N}{\underset{k=1}{\sum }}\alpha_{k}x_{ijk}$$

Explicitly, in terms of our naming convention for variables, the empirical equation for estimation is:
(5)$$\begin{array}{ll} ln\;(IHQ) & =\alpha_{0}+\alpha_{1}{PBR}_{ij}+\alpha_{2}{{PBR}_{ij}}^{2}+\alpha_{3}for\;Profit_{j}+\alpha_{4}\mathrm{Net}\;\mathrm{incom}e_{ij}\\ & +\alpha_{5}Clinical\;services_{j}+\alpha_{6}Length\;of\;stay_{j}+\alpha_{7}Median\;income_{j}\\ & +\alpha_{8}Over65_{j}+\alpha_{9}Patient\;days+\alpha_{10}Research\;volume_{ij}\\ & +\alpha_{11}Research\;impact_{ij}+\alpha_{12}\beta_{j}+\varepsilon_{ij} \end{array}$$
where $\alpha_{k}$ is a vector of parameters representing the impact of the independent variables on the dependent variable, whereas $\beta_{i}$ is a vector of fixed-effects for specializations *j* and $\varepsilon_{ij}$ is the error term.

The dependent variable, Index of Hospital Quality $\left(IHQ_{ij}\right)$ is the 2012–2013 U.S. News & World Report’s “Best Hospitals” score of specialization *i* (*i* = 1, …, 3) in hospital *j* (*j* = 1, …, 50). The *IHQ* provides a composite score between 0 and 100. It is based on Donabedian’s [27] three-element structure-process-outcome paradigm for assessing the quality of healthcare, which is widely used in the literature of healthcare quality estimation (See Di Giorgio et al. for more details [28]). This ranking system is viewed by many physicians and public policy makers as one of the most accurate and influential means of assessing hospital quality [29,30]. (For a more comprehensive review of the development and use of the *IHQ* see the “Best Hospitals” ranking website, Comarow [31]). As a separate proxy for quality of care, we used a sub-dimension of the *IHQ* index, namely, survival rate, a score given in the range 1–10, which refers to a hospital’s success at keeping patients alive. This score is estimated based on in-hospital mortality rates adjusted for case mix (for details, see Chen et al. [29]).

Our main independent variable, $PBR_{ij}\;$(physicians-per-bed ratio) was the number of physicians in specialization *i* in hospital *j* divided by the number of staffed beds and was also obtained from the 2012–2013 U.S. News & World Report.

Two other independent variables, $Research\;volume_{ij}$ and $\;Research\;impact_{ij}$ represented volume and quality, respectively, of biomedical research conducted by hospital physicians in specialization *i* in hospital *j*. We employed a measure of the number of publications for each specialty (see Table 1 for a description of the variable *Research volume*) and H-index [32] as measures of the volume and impact of biomedical research, respectively (for a detailed description of the construction of these variables and a justification for their use, see Tchetchik et al. [25]).

Finally, we collected several relevant covariates known to influence the quality of medical care in hospitals, such as *for-profit* organization dummy, *median income, patient days, over 65,* the number of *clinical services* provided by the hospital, *average length of stay*, and *net income*. These are often used in the literature [29,33] and are described in Table 1 below.

### 2.2. Sample

Our sampling frame was all U.S.-based university hospitals [28]. We employed a stratified sampling approach and identified 50 geographically heterogeneous hospitals (Appendix A lists the sampled hospitals). Within each hospital, we collected secondary data across three specializations: cardiology, oncology and orthopedics. We chose these three specialties because their *IHQ* ranking depends mainly on objective data, namely, structure, process and outcome. Structure refers to resources that relate directly to patient care, outcome relates to risk-adjusted mortality rates, and process was represented by two elements: reputation and patient safety. For the three chosen specializations the weight of the subjective data (reputation) was the lowest. Overall our data comprised 149 department-level observations (one of the hospitals in our sample was not ranked in cardiology by the 2012–2013 U.S. News & World Report which resulted in one missing observation).

Before moving on to the econometric results, we briefly discuss several econometric concerns.

### 2.3. Econometric and Other Methodological Concerns

Simultaneity Bias. Our main concern emerged from the fact that the variable *physicians per specialty* was potentially endogenous as it was determined while taking into account considerations of clinical performance, budget constraints and so on. Hence, the Durbin-Wu-Hausman test was conducted and the null hypothesis that *PBR* is exogenous was rejected, which led us to treat it as an endogenous variable and to pursue an instrumental variable (IV) approach. This requires the assignment of an appropriate instrument, *z*, that should be sufficiently correlated with the endogenous variable *physicians per specialty*, and yet uncorrelated with the error term, *u*, i.e., Cov(z,u)=0. Table 1 presents the instruments employed for *physicians per specialty* and Section 3.1.1 reports on the validity of the instruments and identification tests.

Finally, many of our covariates including patient days, length of stay and net income were plausibly endogenous. These variables were included with a time lag of one year to avoid endogeneity problems.

Other concerns included:

Non-random assignment. As the assignment of physicians to hospitals cannot be assumed to be of a random nature, a suspicion arises that better physicians may be drawn to more reputable hospitals. This phenomenon—the tendency of individuals to match together with others bearing the same characteristics—is known in the economic literature as assortative matching and has been found to exist in labor, marriage and many other markets. The non-random nature of the assignment may pose difficulties concerning measuring the peer effect. To properly isolate the peer effect from the selection effect, one must perform a randomized experiment, which by its nature could not be conducted in our context. Therefore, our results should not be interpreted as reflecting the peer effect but rather as capturing the overall contribution of the physicians-per-bed ratio to clinical performance. Nevertheless, we did control for covariates such as hospital income per physician (with a one-year lag to avoid simultaneity), for-profit/nonprofit, and others in the regressions. We estimated the regressions reported in Table 2 below, separately for two sub-samples, one including hospitals with low revenue per physician and the other including hospitals with relatively high revenue per physician. In both sub-samples, the results with respect to the optimal PBR were similar.

Finally, there was a challenge that we had to address with respect to our main independent variable, the adjusted number of physicians in each specialty (defined as the number of physicians in each specialty divided by the number of staffed beds). As mentioned above, figures on the physician workforce were obtained from the U.S. News & World Report. These figures are detailed to the level of the specialty, yet there is no distinction between part- and full-time employees, an important distinction if one wants to compare the link between workforce and healthcare quality across departments and hospitals (other databases, such as the Annual Hospital Association (AHA) and the Healthcare Cost and Utilization Project (HCUP) also do not distinguish between part- and full-time employees). Thus, to obtain a measure of full-time employees, for each of the 149 departments we selected from the U.S. News & World Report for 2012–2013, only those physicians whose affiliation status listed only one hospital (see Appendix A for example). For physicians who had only one affiliation we further checked their list of publications since 2013 and verified that in each article the physician appeared solely under the affiliation of that hospital. In the analysis that follows, we employed both figures, total physicians and full-time employees.

## 3. Econometric Results

### 3.1. Hypothesis Testing

#### 3.1.1. Hypothesis 1: Main Effects

Table 2 below reports the results of two IV regressions with ln (*IHQ*) as the dependent variable. The regression reported in Model (1) employed the variable PBR as the main independent variable and the regression reported in Model (2) employed the variable PBR_FTEs (see Table 1 for the differences between PBR and PBR_FTEs).

Due to the endogeneity of the physicians-per-bed ratio as discussed in Section 2.3, an instrumental variables method was employed to estimate both models. We employed Stata’s 15.1 ivreg2 command which accommodates a wide range of single-equation estimation methods for the linear regression model, including two-stage least squares, the generalized method of moments (GMM) and limited-information maximum likelihood [34,35]. The IV-GMM estimation was selected as it can improve upon the traditional two-stage least squares (2SLS) approach [36]. Note that the Wu-Hausman F, and Durbin-Wu-Hausman square, tests for endogeneity of the number of physicians per specialty for total number of physicians and for full-time physicians rejected the null of exogeneity (*p* < 0.0001).

Table 2 provides evidence for the existence of an inverted U-shaped relationship between department size, defined as PBR, and the quality of healthcare (Hypothesis 1). In both specifications in Table 2, the linear and quadratic coefficients of PBR (PBR_FTEs) are significantly positive and negative, respectively, at the 1% level. These findings alone are insufficient to determine the existence of an inverted-U, as a significant quadratic term can identify an inverted-U as well as lesser degrees of curvilinearity in relationships that are monotonic within the data range. Therefore, an exact test for the presence of an inverse U-shaped relationship between PBR (PBR_FTEs) and ln (*IHQ*) over an interval, was conducted using Stata’s 15.1 Utest [37] and was confirmed at the 1% level for both models in Table 2.

The optimal *PBR* that maximizes clinical performance was calculated by taking the derivative of the empirical specification (Equation (5)) with respect to *PBR*, i.e., $\frac{\alpha_{1}}{-\left(2\cdot \alpha_{2}\right)}$. The optimal ratio for Model (1) which employs the variable PBR was 0.36, whereas the optimal ratio for Model (2) which employed the variable PBR_FTEs was 0.23.

With respect to the control variables, in both models, hospital net income, the number of clinical services and number of patient days were positively correlated with *IHQ* with very similar coefficients across both regressions.

The adjusted R^2^ for Models (1) and (2) was 0.37 and 0.32 respectively, which was sufficiently high overall. The results of the regressions are reassuring considering the relatively small sample size and its cross-sectional nature. 

Partial Elasticities. Calculated at the sample’s mean point, an increase of 1% in *PBR* was associated with a rise of 0.17% in *IHQ* in Model (1) and of 0.23% in Model (2).

The existence of an inverted-U relationship justified the estimation of these elasticities at the two tails, and not just the sample mean point. We found that for Model (1) a 1% increase in PBR for the hospital with the smallest such ratio (left tail) was associated with an 82.4% increase in its clinical performance, whereas a 1% decrease in PBR for the hospital with the highest ratio raised its clinical performance by 54.2%. In Model (2) that accounts for FTE’ physicians only, a 1% increase in PBR for the hospital with the smallest such ratio was associated with a 120.9% increase in its clinical performance, whereas a 1% decrease in PBR for the hospital with the highest ratio raised its clinical performance by 105.6%.

Validity of instruments and identification. The under/weak identification tests (which determine whether the excluded instruments are not or are weakly correlated with the endogenous regressors) were rejected, indicating that the model was identified and there was no issue of weak identification. Finally, testing over-identification was not relevant as our model was exactly identified.

#### 3.1.2. Hypothesis 1: Interaction Effects

As Table 1 demonstrates, there are pronounced differences in the PBR/PBR_FTEs ranges between the three specialties. For that reason, we next tested whether and to what extent the optimal PBR differed by specialty. Employing the first stage results of Models (1) and (2) in Table 2, fitted values for PBR and PBR_FTEs were calculated and employed to calculate the interaction terms between *PBR*/PBR_FTEs and their quadratic forms, and each specialty dummy variable. Table 3 reports the results of these regressions. We employed robust standard errors in both regressions as White’s test for heteroskedasticity rejected the null of homoskedasticity.

As Models (1) and (2) in Table 3 reveal, the inclusion of interactions improved the explanatory power of the model as reflected by the increased adjusted R^2^ (0.59 compared to 0.37 in Table 2 for PBR, and 0.42 compared to 0.32 in Table 2 for PBR_FTEs). It appears that the optimal *PBR* differs across specialties. See Figure 1 for illustration of the results for Model (1). Finally, we calculated the differential partial elasticities (at the sample’s means). We found that a one percentage point increase in *PBR* was associated with a 0.27%, 0.13% and 0.30% increase in *IHQ*, in oncology, orthopedics and cardiology in Model (1), respectively, and 0.29%, 0.16% and 0.20% in Model (2), respectively. Figure 1, illustrates the observed and fitted values of ln(*IHQ*) as a function of *PBR* (left panel) and PBR_FTEs (right panel) according to the results of Model (1).

Note that the presence of several outliers in Figure 1 (marked in yellow) raises a concern over the possibility that the curvilinear relationship may in fact be driven by these outliers. Indeed, while the graphs are bivariate, and some other variables may be in play, we nevertheless repeated both figures without the suspicious observations and the quadratic pattern of the relationships between PBR/PBR_FTEs and ln(*IHQ*) remained stable (see Appendix A). As described in Section 3.2, we also conducted a robustness check, namely Cook’s distance, to estimate the influence of possible outlier data points, which sustained our conclusion of non-linearity.

#### 3.1.3. Hypothesis 2: The Effect of Research Impact on the Relationship between PBR and Healthcare Quality

Based on previous studies [25,26] that found a positive effect of research impact on healthcare quality, we included in the IV-GMM regressions analysis reported in Table 2 the research impact and research volume variables. Table 4 provides the results of these regressions for both PBR and PBR_FTEs (columns 1 and 2, respectively).

As in Table 2, both models in Table 4 provide evidence for the existence of an inverted U-shaped relationship between PBR/PBR_FTEs and the quality of healthcare (which supports Hypothesis 1). The coefficients of the control variable are also very similar to the coefficients in Table 2.

In line with previous studies [25,26] Models (1) and (2) indicate that the effect of research impact on healthcare quality was positive and significant, while the effect of research volume was not significant. Moreover, the adjusted R^2^ increased when research impact was included from 0.37 (for PBR) and 0.32 (for PBR_FTEs) in Table 2, to 0.38 and to 0.39 in Table 4, respectively. In both models we can see that when research impact is accounted for, the effect of PBR (or PBR_FTEs) on *IHQ* declined which is reflected in both the partial elasticity and the optimal PBR. For example, an increase of 1% in *PBR* was associated with a rise to 0.06% in *IHQ* in Model (1) compare to 0.17% in Table 2, when research impact was not accounted for. In fact, the calculated elasticity of research impact in Model (1) in Table 4 is 1.19%, which is twice as large as the elasticity of *PBR* in the same model. Finally, the optimal PBR/PBR_FTEs) decreases from 0.36 (for PBR) and 0.23 (for PBR_FTEs) in Table 2 to 0.32 and 0.21 in Table 4, respectively. These results raise a possibility that *Research impact* is a mediator for the relationship between PBR/PBR_FTEs and *IHQ.* A priori, there is no reason to assume that research impact functions as a mediator, i.e., that it accounts for the relation between PBR/PBR_FTEs and *IHQ.* Correlation tests also revealed a relatively low correlation of 0.28 between research impact and PBR and a weakly significant correlation (*p* < 0.077) of 0.14 between research impact and PBR_FTEs. These correlations probably result from the correlations between the number of physicians in the ward (the nominator of PBR) and its research impact.

However, in line with Hypothesis 2, we expected research impact to be a moderator variable that influences the strength of the relationship between PBR/ PBR_FTEs and *IHQ*. To test for a moderation effect, we estimated two additional regressions in which PBR/PBR_FTEs were interacted with research impact as measured by the H-index (see Appendix A for these regression results). The coefficients of PBR * H-index and PBR_FTEs * H-index were positive and significant, providing support for our second hypothesis. Research quality strengthened the effect of adding a single physician on healthcare results. Figure 2 presents the predicted *IHQ* as a function of PBR resulting from this regression (Model 1 in Appendix A). As can be seen, when research impact is interacted with PBR, the optimal PBR decreases.

### 3.2. Robustness Checks for Hypotheses 1 and 2

*Hypothesis 1:* First, as Figure 1 demonstrates, the curvilinear relationship appears to be driven by the presence of the outliers marked in yellow, in both cardiology and orthopedics. To test for the possible influence of individual data points we employed Cook’s distance (using Stata’s 15.1 Cooksd postestimation option [35]). Several suspicious observations, for which Cook’s distance was greater than n/4, were identified. Nevertheless, excluding these observations from the regression did not affect the stability of the model and the quadratic relationships between PBR/PBR_FTEs and *IHQ* remained.

Second, in order to examine whether a non-linear relationship still held when other performance measures were taken into account, we conducted additional analyses with alternative dependent variables (namely, survival rate, hospital net income, hospital revenues and annual average amount of NIH received over the years 2006–2012.). Survival rate is a sub-dimension of the *IHQ* index, which refers to hospital’s success at keeping patients alive. The results for the dependent variable Survival rate are reported in Appendix A. They are qualitatively similar to those reported in Table 4. Stata’s 15.1 Utest confirmed the existence of an inverted U-shaped relationship with respect to both PBR and PBR_FTEs at the 5% level, with optimal ratios of 0.41 and 0.23, respectively.

Appendix A reports the results of the regressions referring to dependent variables: hospital net income, hospital revenues and annual average amount of NIH received over the years 2006–2012 for the independent variable PBR. It appears that there exists an optimal *PBR* for these performance measures; 0.41 and 0.42 for maximizing hospital revenues and net income, respectively, and 0.6 for maximizing NIH funding.

*Hypothesis 2*: We tested Model (1) in Table 4 using another measure of research impact, namely, average citations per physician, averaged for each specialty in each hospital (for more details on the construction of this variable see Tchetchik et al. [25]). The results were similar to those reported in Table 4, with the coefficient of optimal average citations per physician being positive and significant and the calculated PBR equal to 0.33. Due to space limitations, the results of this regression are not presented.

## 4. Discussion

This research contributes conceptually and empirically to the literature on the relationship between costs and quality of healthcare in the hospital sector. Despite a relatively large amount of evidence available on this topic, this remains an issue of interest which deserves further investigation. While most previous studies used aggregate measures (states or regions) of costs and medical outcomes, here we take a disaggregated approach that allows us to take heterogeneity among hospitals and between specialties into account.

Healthcare costs in the U.S. represent a sizeable share of its GDP (over 17% according to a 2014 OECD report [37]. According to a comprehensive report published by the Commonwealth Fund in 2014, healthcare spending is expected to grow by an average of 5.6 percent annually over the next decade, a much higher rate than other sectors in the economy. Yet, based on outcome indicators such as quality (in terms of effectiveness, safety, coordination, and patient-centeredness) the performance of American healthcare is lacking. According to a 2017 report of the OECD, in terms of inefficiency the U.S. is not alone “About fifth of the expenditure devoted to healthcare in the OECD countries makes very little, if any, contribution to health outcomes” [38]. There are several identified sources for inefficiencies including administrative hassles, timely access to records and test results among others. By focusing on the link between direct input (labor instead of costs) and healthcare outcomes at the micro-level (the ward level instead of national, state or provincial levels) this research provides evidence that the relationships between expenditure and outcome are not linear.

Our empirical analyses provided strong support for both of our hypotheses. We established the existence of an inverted U-shaped relationship between the physicians-per-bed ratio at the ward level and the ward level *IHQ*. While our paper is not the first to find an inverted U-shaped relationship (other works found an inverted U-shape relationship between overall costs and quality, mostly in nursing homes [16,39,40]) what makes our approach unique is the focus on the ward-level relationships between labor and capital inputs (e.g., number of beds) and clinical performance. This approach is warranted because scale effects due to spillovers should be measured within elementary micro-units. Therefore, while our approach does not reveal the overall link between healthcare spending and outcomes, it does allow for better measurement of an important derivative of this relationship. The existence of an interior optimal PBR reaffirms our hypothesis that organizational limitations pose natural boundaries to increasing returns. Moreover, our empirical investigation reveals that although positive and significant, the partial elasticity of healthcare quality with respect to *PBR* is significantly smaller than the contribution of other factors, specifically the quality of the research conducted by the ward, a point which has important policy implications.

In support of our second hypothesis, we showed that research quality moderates the relationship between healthcare quality and the physicians-per-bed ratio. When interaction with research impact was included, the optimal PBR fell substantially. We confirmed that as a general rule, quality speaks louder than quantity; i.e., the effect of increasing research quality as measured by the average H-index was two-times greater than the effect of raising the physicians-per-bed ratio. A direct implication of this result is that raising the quality of the medical workforce results in a lower optimal physicians-per-bed ratio needed to sustain given healthcare results. We believe that this result supports the existence of what appears to be a ‘quantity-quality trade-off’, where greater research quality may reduce the workforce size needed to sustain given healthcare results. This bears important policy implications. For example, raising the quality of the workforce (both medical and non-medical) through improvements in hiring selection procedures, training, and the imposition of better coordination and management protocols may prove a more cost-effective way to achieve better healthcare results than simply increasing the workforce. This fact should not come as a surprise; ever since the seminal contribution of Solow—a Nobel Prize laureate in Economics who discovered the “Solow Residual”—the fact that improvements in the methods of production make a considerable contribution to the growth of output, over and above the mere increase in labor and capital inputs has been acknowledged. Better organization of the ward-level environment could therefore pay huge dividends where improving healthcare outcomes is concerned.

## 5. Conclusions

Studying the link between healthcare spending and outcomes is highly complicated. In this paper, instead of studying the link between overall costs and outcomes, we focused on the relationship between the physicians-per-bed ratio and healthcare results, to trace the existence of a congestion effect. We relied on departments’ cross-sectional diversity in both the physicians-per-bed ratio and healthcare results, to establish the existence of interior optimal PBR, while controlling for other relevant covariates. We demonstrated that an interior optimal physicians-per-bed ratio might vary across specialties. This implies that each specialty might have its own characteristics and unique features, and hence may support a better reallocation of resources among different units within a hospital.

On balance, the marginal effect on healthcare results of adding another physician, while positive and significant, appears to be less important than the effect of raising ward-level research quality. Moreover, our results demonstrate an important allocative principle—a substitution effect between ‘quantity’ and ‘quality’. If one accepts the claim that there is a synergy between research and practice, so that exceling in biomedical research goes hand in hand with being a better physician, then it comes as no surprise that quality substitutes for quantity. We believe that further empirical investigations are needed to support this claim, specifically studies that use panel data to control for omitted variables bias, as well as to capture the dynamic nature of the data. (Except for Di Giorgio et al. [27] most related studies have relied on cross-sectional analyses.). Yet, we believe that our main results are in line with many other studies, some of which were mentioned in this text, that call for a better utilization of existing resources, rather than adding new resources to the healthcare system.

Note that while our empirical results yield optimal PBRs, they should not be interpreted as policy recommendations in a quantitative sense per se. The message delivered by our results is more qualitative than quantitative in nature; our main point is that there exists an interior, optimal PBR which is ward-/specialty-specific such that expanding the workforce might not always lead to better results. Moreover, enhancing the research opportunities and excellence of the medical workforce may carry much greater effect on healthcare outcomes than simply adding more staff members.

Finally, while we controlled for relevant covariates, we acknowledge the potential influence of other hospital labor inputs (e.g., nurses, paramedics as well as non-clinical staff) on healthcare outcomes. Unfortunately, these data were not available for our sample period of 2012–2013 at the ward level. Further studies addressing this topic are encouraged to make an effort to include these variables in their analyses.

## Figures and Tables

**Figure 1 ijerph-16-00761-f001:**
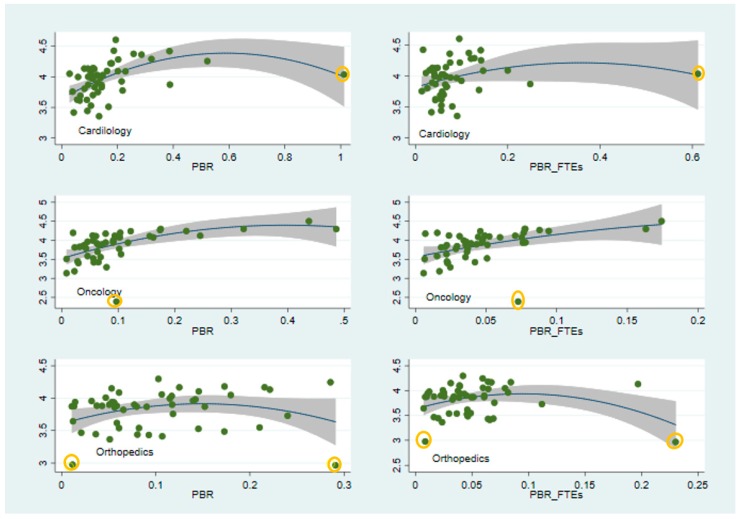
Predicted vs. observed values of ln(*IHQ*) as a function of PBR and PBR_FTEs. Predicted ln(*IHQ*) are in solid black lines. Observed ln(*IHQ*) are in green dots. Shaded areas represent 95% confidence intervals. All other variables are held at means.

**Figure 2 ijerph-16-00761-f002:**
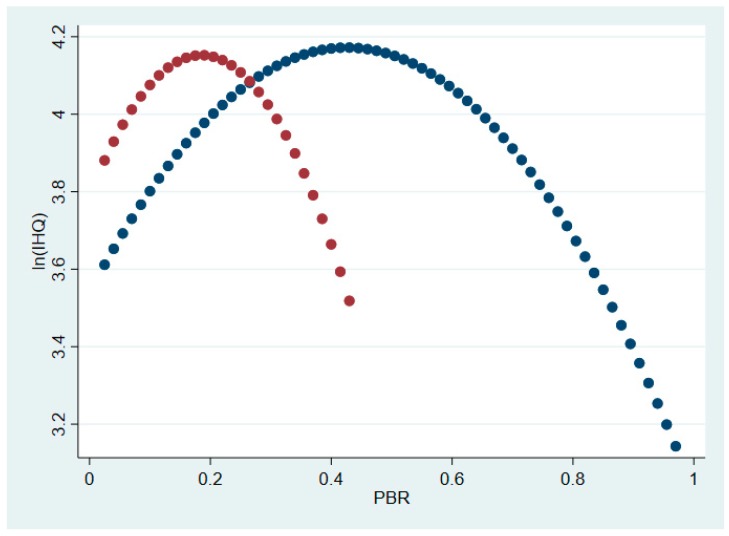
Predicted values of ln(*IHQ*) as a function of PBR for different values of research impact. The blue dots represent the predicted values of ln(*IHQ*) as a function of PBR where the H-index is held at its modal value (0.5). The red dots represent the predicted values of ln(*IHQ*) as a function of PBR where the H-index is held at its second most frequent value (2.5).

**Table 1 ijerph-16-00761-t001:** Descriptive statistics of variables employed in the regression analysis.

Variable	Description	Descriptive Statistics *
**Dependent Variables**		
*IHQ* ^(1)^	Department’s *IHQ*, in 2012–2013 expressed as a composite score between 0 and 100	50.81 (15.08) Min–Max: 10.90–100.0
Survival ^(1)^	Department’s patient survival rate—a sub-dimension of *IHQ*, for 2012–2013	7.71 (1.70) Min–Max: 3–10
**Independent Variables**		
Research impact ^(2)^	Median H-index for each department	3.18 (2.40) Min-Max: 0.5–13.5
Research volume ^(3)^	Median number of publications per physician (FTE) per annum for each department starting from the year his/her first paper was published	1.50 (0.86) Min–Max: 0.18–4.38
PBR ^(1)^	Number of physicians in each specialty divided by the number of staffed beds	Oncology 0.10 (0.09) Min–Max: 0.008–0.48 Cardiology: 0.17 (0.16) Min–Max: 0.03–1.01 Orthopedics: 0.10 (0.07) Min–Max: 0.007–0.23
PBR_FTEs ^(1,4)^	Number of full-time physicians in each specialty divided by the number of staffed beds	Oncology 0.047 (0.03) Min–Max: 0.006–0.17 Cardiology: 0.08 (0.09) Min–Max: 0.014–0.612 Orthopedics: 0.05 (0.04) Min–Max: 0.01–0.23
For-Profit ^(5)^	=1 if the hospital is a for-profit organization	0.30
Clinical services ^(5)^	Number of clinical services provided by the hospital	35.03 (5.17) Min–Max: 21.00–44.00
Length of stay ^(5)^	Average number of days of hospitalization (with a time lag of one year)	6.27 (0.80) Min–Max: 4.43–7.92
Over 65 ^(6)^	Share of population over 65 years old in the 3 geographically closest zip codes to the hospital	0.13 (0.03) Min–Max: 0.08–0.19
Median income ^(5)^	Median income of three geographically closest zip codes to the hospital, in current thousands of USD	40.87 (13.25) Min–Max: 22.95–83.58
Net income ^(5)^	Hospital’s net income/loss per bed in current hundred thousand of US dollars (time lag of one year)	1.09 (0.97) Min–Max: −0.69–4.03
Patient days ^(5)^	The total number of patient days in hundred thousand (with a time lag of one year)	20.05 (11.42) Min–Max: 4.10–74.12
**Instruments**		
Total physicians ^(1)^ per bed	Total number of physicians per total beds in each hospital	1.806 (1.39) Min–Max: 0.10–7.769

* Means are followed by standard deviations in parentheses and minimum-maximum range (for non-dummy variables). Sources: ^(1)^ 2012–2013 U.S. News & World Report’s “Best Hospital” ranking; ^(2)^ Web of Science database, data collected during 2012–2013; ^(3)^ PubMed database, data collected during 2012–2013; ^(4)^ Last paragraph of Section 2.3 describes the method used to obtain FTE’s (FTE: full time employee) out of the total number provided by U.S. News & World Report (source 1); ^(5)^
http://www.ahd.com/freesearch.php; ^(6)^ United States Census Bureau.

**Table 2 ijerph-16-00761-t002:** IV–GMM regression. Dependent variable: Ln (*IHQ*), endogenous variable: PBR.

		Model (1)	Model (2)
PBR	2.481 ***	PBR_FTEs	5.283 ***
	(3.27)		(3.05)
PBR^2^	−3.439 ***	(PBR_FTEs)^2^	−11.477 ***
	(−3.04)		(−3.1)
Orthopedics ^(1)^	−0.057		−0.048
	(−1.09)		(−0.88)
Cardiology ^(1)^	0.042		0.037
	(0.71)		(0.58)
For-Profit	0.005		0.040
	(0.08)		(0.67)
Length of stay	−0.054		−0.051
	(−1.53)		(-1.38)
ln (Median income)	−0.079		−0.130
	(−0.97)		(−1.48)
Over 65	0.312		0.371
	(0.36)		(0.41)
Patients days	0.008 ***		0.010 ***
	(3.05)		(3.74)
Clinical services	0.015 ***		0.016 ***
	(2.71)		(2.75)
Net Income	0.091 ***		0.092 ***
	(3.08)		(3.04)
Constant	4.022 ***		4.422 ***
	(4.59)		(4.77)
Adj-R^2^	0.37		0.32
*N*	149		149
Optimal PBR	0.36	Optimal PBR_FTEs	0.23

Notes: *z* statistics in parentheses, *** *p* < 0.01. ^(1)^ The omitted specialty is oncology.

**Table 3 ijerph-16-00761-t003:** Regression analysis including interaction effects of the physicians-per-bed ratio with the different specialties. Dependent variable: Ln (*IHQ*).

	Model (1)		Model (2)
PBR ^(1,2)^	5.291 ***	PBR_FTEs ^(1,2)^	7.011 ***
	(3.46)		(3.42)
PBR^2 (1,2)^	−14.890 ***	PBR_FTEs ^(1,2)^	−29.470 ***
	(−3.79)		(−4.71)
PBR * # Ortho	−2.980 ***	PBR_FTEs * Ortho	−3.486 ***
	(−4.10)		(−2.84)
PBR^2^ * Ortho	8.913 ***	(PBR_FTEs)^2^ * Ortho	16.45 ***
	(5.26)		(4.85)
PBR * Cardio	−3.325 **	PBR_FTEs * Cardio	−4.676 ***
	(−2.50)		(−3.47)
PBR^2^ * Cardio	13.950 ***	(PBR_FTEs)^2^ * Cardio	25.88 ***
	(3.95)		(8.21)
Orthopedics	0.081		0.043
	(0.64)		(0.72)
Cardiology	0.106 *		0.109
	(1.95)		(1.14)
For-Profit	−0.040		0.021
	(−0.55)		(0.28)
Length of stay	−0.036		−0.029
	(−0.99)		(−0.85)
ln (Median income)	−0.060		−0.0820
	(−0.71)		(−1.03)
Over 65	−0.834		−0.440
	(−0.75)		(−0.39)
Patients days	0.001		0.004
	(0.44)		(0.96)
Clinical services	0.012		0.009
	(1.28)		(0.91)
Net income	4.250 ***		4.950 ***
	(2.94)		(3.51)
Constant	−8.970 *		−10.84 **
	(−2.00)		(−2.62)
Adj-R^2^	0.59		0.44
*N*	149		149
Optimal PBR	Oncology 0.18	Optimal PBR_FTEs	Oncology 0.12
	Orthopedics 0.19		Orthopedics 0.13
	Cardiology 0.39		Cardiology 0.32

Notes: *t* statistics in parentheses, * *p* < 0.10, ** *p* < 0.05, *** *p* < 0.01. ^(1)^ PBR and PBR_FTEs were calculated from first-stage results of the IV regression in Model 1 and Model 2, respectively. ^(2)^ These coefficients refer to the effect of PBR/PBR_FTEs on oncology, the omitted specialty.

**Table 4 ijerph-16-00761-t004:** IV-GMM–regression. Dependent variable: Ln(*IHQ*), endogenous variable: PBR.

	Model (1)		Model (2)
PBR	1.808 **	PBR_FTEs	4.036 **
	(2.36)		(2.06)
PBR	−2.782 **	(PBR_FTEs)^2^	−9.391 **
	(−2.54)		(−2.37)
Orthopedics ^(1)^	−0.011		−0.007
	(−0.20)		(−0.13)
Cardiology ^(1)^	0.087		0.077
	(1.40)		(1.13)
For-Profit	0.030		0.053
	(0.56)		(0.94)
Length of stay	−0.060 *		−0.057
	(−1.78)		(−1.64)
ln(median income)	−0.029		−0.075
	(−0.36)		(−0.83)
Over 65	0.529		0.535
	(0.63)		(0.61)
Patients days	0.008 ***		0.009 **
	(3.01)		(3.51)
Clinical services	0.014 ***		0.015 ***
	(2.65)		(2.73)
Net Income	0.091 ***		0.090 ***
	(3.26)		(3.18)
Research impact	0.029 **		0.025 *
	(2.43)		(1.86)
Research volume	0.019		0.020
	(0.49)		(0.53)
Constant	3.474 ***		3.852 ***
	(3.97)		(4.08)
R^2^	0.38		0.39
*N*	149		149
Optimal PBR	0.32	Optimal PBR_FTEs	0.21

Notes: *z* statistics in parentheses. * *p* < 0.10, ** *p* < 0.05, *** *p* < 0.01. ^(1)^ The omitted specialty is oncology.

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
