# Peer review of "An Empirical Investigation of “Physician Congestion” in U.S. University Hospitals"

_ijerph, 2019, doi:10.3390/ijerph16050761_

Reviewer 1 Report

Title: An Empirical Investigation of 'Physician Congestion' in U.S. University Hospitals

Comment 1

Page 1, row: 39-41: Authors wrote that “Considering ever-increasing expenditure, the relationship between healthcare spending and outcomes is a topic of keen interest stimulating heated public discourse among proponents of more healthcare spending and supporters of "bending the cost curve".

Authors should make clear that outcome of what? 

Comment 2

Since the values of dependent variables range between 0% and 100%, authors might have transformed them into the logit function, which is the inverse of the logistic transform. I am not an expert in statistics, but suggest the authors to reconsider the logistic transformation. IHQ is a with any score of 1 or less would get negative values if transfers into LN. 

Comment 3

The authors focused on “number of physicians” as the main explanatory variable while considering other covariates in the models. However, the regression models have not controlled for variations in other staff sizes (nurses, paramedics as well as non-clinical) since healthcare outcome is dependent on case-mix, not just number of physicians. 

I suggest the authors to include the ‘number of other staffs’ in the models.  This is important to address.

Comment 4

Page 14, row 416: Authors mentioned that “Despite a relatively large amount of evidence available on this topic, this remains an issue of interest which deserves further investigation.” 

But the authors did not elaborate what further investigation is deserved and why? 

Comment 5

Page 14, row 431-433: Authors mentioned that “While our paper is not the first to find an Inverted-U relationship what makes our approach unique, is the focus on a narrower dimension of expenditure, i.e. expenditure on the physician workforce (as opposed to overall expenditure), and its association with clinical performance.

I find that the authors tested the relationship between the number of physicians and treatment outcomes, but did not put the expenditure related to physicians in the analyses. This claim (above), thus, was not supported by this research. I would expect different wage size of the physicians in different states and the number of physicians may not fully correspond to the expenditures of physicians.  

Comment 6

What contributions this paper could make in the knowledge hub should be specified. The application of this research in the context of policy should be elaborated and 

Author Response

Comment 1

Page 1, row: 39-41: Authors wrote that “Considering ever-increasing expenditure, the relationship between healthcare spending and outcomes is a topic of keen interest stimulating heated public discourse among proponents of more healthcare spending and supporters of "bending the cost curve".

Authors should make clear that outcome of what? 

RE: We wish to thank the reviewer for calling our attention to this disclarity: we now rephrase the sentence and use ‘clinical performance’ instead of ‘outcome’ (see P1, L39 in the revised MS).

Comment 2

Since the values of the dependent variables range between 0% and 100%, authors might have transformed them into the logit function, which is the inverse of the logistic transform. I am not an expert in statistics, but suggest the authors to reconsider the logistic transformation. IHQ is a with any score of 1 or less would get negative values if transfers into LN. 

RE: This is a good point. Note however that when transforming a variable that accepts values in  into a Logarithmic form, one indeed need to be concerned over the change of sign. That said, our dependent variable takes values in (0,100], hence this concern is dismissed.

 Comment 3

The authors focused on “number of physicians” as the main explanatory variable while considering other covariates in the models. However, the regression models have not controlled for variations in other staff sizes (nurses, paramedics as well as non-clinical) since healthcare outcome is dependent on case-mix, not just number of physicians. 

I suggest the authors to include the ‘number of other staffs’ in the models.  This is important to address.

RE: We thank the reviewer for this very good comment. Unfortunately, data on the “variations in other staff sizes” at the specialty level and in the resolution required (i.e. nurses, paramedics and non-clinical) for the sample period 2012-13 is not available to the best of our knowledge. The Healthcare Cost and Utilization Project (HCUP) obtained some limited information on nurse staffing only and included it for several years of the Nationwide Inpatient Sample (NIS). The 2007-2011 NIS contains full time equivalent (FTE) staffing of registered nurses, licensed practical nurses, and nurse aides. Yet, there is no data for the years 2012-2013. Future studies addressing this topic should make efforts to obtain these data and include it in the analysis. We address this issue in the revised MS in the Discussion section (P15, L516-520)   

Comment 4

Page 14, row 416: Authors mentioned that “Despite a relatively large amount of evidence available on this topic, this remains an issue of interest which deserves further investigation.” 

But the authors did not elaborate what further investigation is deserved and why? 

RE: We appreciate this comment. To make this point clearer, we rephrase that sentence as follows: "Despite a relatively large amount of evidence available on this topic, this remains an issue of interest which deserves further investigation; while most previous studies used aggerate measures (sates or regions) of costs and medical outcomes, here we take a dis-aggregated approach that allows us to take heterogeneity among hospitals and between specialties into account. (see P13, L444-446 in the revised MS)

Comment 5

Page 14, row 431-433: Authors mentioned that “While our paper is not the first to find an Inverted-U relationship what makes our approach unique, is the focus on a narrower dimension of expenditure, i.e. expenditure on the physician workforce (as opposed to overall expenditure), and its association with clinical performance.

I find that the authors tested the relationship between the number of physicians and treatment outcomes, but did not put the expenditure related to physicians in the analyses. This claim (above), thus, was not supported by this research. I would expect different wage size of the physicians in different states and the number of physicians may not fully correspond to the expenditures of physicians.  

RE: We wish to thank the reviewer for this important comment. Unlike most previous studies that correlated aggregate measures of costs and clinical outcomes, here we wanted to test the relationships between resources and outcomes within micro-units (ward level). With a lack of detailed information regarding physicians waged in the ward level, we used an input-output approach where resources are represented by capital and labor inputs and output is the clinical performance. That said, we were not clear enough on this point in the original manuscript, and this is now taken care of in the revised manuscript.

It is now stated that “While our paper is not the first to find an Inverted-U relationship what makes our approach unique, is the focus on the ward level relationships between labor and capital inputs (e.g. number of beds) and clinical performance”. See P14, L463-464 in the revised MS.

 Comment 6

What contributions this paper could make in the knowledge hub should be specified. The application of this research in the context of policy should be elaborated and 

RE: We now better emphasize the contribution of out paper and elaborate on its policy context (See P. 14 L483-491 in the revised version): “For example, raising the quality of the workforce (both medical and non-medical) through improvements in hiring selection procedures, training, and the imposition of better coordination and management protocols may prove a more cost-effective way to achieve better healthcare results than simply adding more workforce. This fact should not come as a surprise; ever since the seminal contribution of Robert Solow, a Nobel-prize laureate in economics – who discovered the “Solow Residual” – we acknowledge the fact that improvements in the methods of production have a considerable contribution to the growth of output, over and above the mere increase in labor and capital inputs. Better organization of the ward-level environment could therefore pay huge dividends when improving healthcare outcomes is concerned.”

Reviewer 2 Report

Independent of the basic idea, mathematical and bio-statistical methods are extremely precise and leave no doubt about the correct nature of the results achieved. However the approach of the core idea raises some questions.
Page 2, 91-92: missing non-linearity is not a convincing evidence for neglecting this relationship in previous studies which examined the physician per bed ratio only within reasonable limits of managing human resources.
Page 3, 135-136: research quality is by this approach an umbrella definition in terms of theoretical and applied sciences. By all probability, MDs in hospitals were engaged in studies about applied medical sciences. Concerning theoretical studies, there is no direct relationship with caring for in-patient population.

Author Response

Replies to reviewer’s 2 Comments and Suggestions for Authors:

Independent of the basic idea, mathematical and bio-statistical methods are extremely precise and leave no doubt about the correct nature of the results achieved. However the approach of the core idea raises some questions.
Page 2, 91-92: missing non-linearity is not a convincing evidence for neglecting this relationship in previous studies which examined the physician per bed ratio only within reasonable limits of managing human resources

RE:  We wish to thank the reviewer for bringing it up. We certainly acknowledged the fact that previous studies might have overlooked the possible existence of non-linearities owing to limited scale of the main independent variable (i.e. PBR in our study). In the revised manuscript we added this to the text in P2, L92.

Page 3, 135-136: research quality is by this approach an umbrella definition in terms of theoretical and applied sciences. By all probability, MDs in hospitals were engaged in studies about applied medical sciences. Concerning theoretical studies, there is no direct relationship with caring for in-patient populationי...

RE: This is a very good point, we added the following text as note: “while research quality refers both to theoretical and applied research, it seems likely that in the context of physicians in hospitals it is mainly applied research that directly contributes to clinical performance.” (P3, L137-139)

Reviewer 3 Report

IJERPH-411427

An Empirical Investigation of 'Physician Congestion' in U.S. University Hospitals

Resubmit with Major Revisions IF further data analysis can increase the credibility of the core results.

The authors have assembled a large dataset, for which they should be congratulated.

The core claim of the paper is that there is an optimum ratio of physician-to-bed (PBR) staffing. The authors suggest some plausible reasons why this might be expected, and fit curves to data to suggest it is indeed seen in the data.

The core relationship tested is Equation 6, the relationship between log (quality) and a quadratic in PBR.

I have two related major issues that lead me to find this central analysis unconvincing:

                     i.            The implications of the values of PBR (or PBR_FTEs), the key variable:

a.       Table 1 shows that it, has an extreme range (0.008 – 1.01 (or 0.006 – 0.61): order of magnitude difference of over 100!) SO are these really a homogeneous (comparable) set of hospitals? The Cardiology graph in Figure 1 shows this in that specialty, with an outlier over 1 twice about twice as large as any other. Is this the same organisation as the one with the highest PBRs in the two top graphs – where this highest PBR hospital has unusually low quality?

b.      The Oncology graph in Figure 1 suggests there are some negative PBRs. What does this mean?!

                   ii.            The fit of the quadratic in PBR:

a.       Although the regression stats (Table 3) show the quadratic term is very statistically, the graphs (Fig 1) make these fits look very dubious. The worst is Cardiology, where the curvilinear relationship appears to be driven entirely by the one outlier (noted in (i). The other two curves also look very sensitive to what look like outliers: bottom right in particular, but maybe also bottom left - (so 1 or 2 hospitals with unusually low quality metrics). In these top two graphs, are the 95% CIs not consistent with failing to reject a linear fit?

b.      Are these potentially influential points in the top 2 or all 3 graphs actually the same 2 hospitals?

Of course the graphs are bivariate, so there are some others variables in play, and the authors are to be congratulated for showing the graphical visualisation of fits and their confidence intervals. These are important and particularly revealing here rather than just blind tables of regression output - but the authors should have paid heed to the weaknesses the graphs show.

My view is that these issues seriously undermine the credibility of the analysis and so findings.

(i) needs consideration and explanation.

(ii) Needs a rethink. The authors should (anyway) consider conventional multiple regression robustness diagnostics for their models. Here, in particular, the influence of individual datapoints could be considered: like the leverage, Cook’s distance …

 Minor issues:

·         The paper is carelessly written – these are hygiene factors!

o   Typos (e.g. “weekly” for weakly, “sub-sect 2.1” should be 2.2, p.5 line 188 j should be i etc)

o   Citation style is a mix of Vancouver and Harvard style-families

o   Some refs are wrong (e.g. 31)

o   The description of physician intensity is ambiguous in many places: it talks about volumes or numbers of physicians and size of dept I think the ratio is meant

o   The annual average of publications is ambiguous in the text: average over staff? FTEs?

o   Footnotes should be integrated into the text

o   Formatting is erratic in places (p. 7 section headings)

o   The regressions seem to give the sample size as 149, the dataset is said to contain 150 points, but there is no comment about missing data.

Author Response

Reply to reviewer’s 3 comments and suggestions:

The authors have assembled a large dataset, for which they should be congratulated.

The core claim of the paper is that there is an optimum ratio of physician-to-bed (PBR) staffing. The authors suggest some plausible reasons why this might be expected, and fit curves to data to suggest it is indeed seen in the data. The core relationship tested is Equation 6, the relationship between log (quality) and a quadratic in PBR.

I have two related major issues that lead me to find this central analysis unconvincing:

                     i.   The implications of the values of PBR (or PBR_FTEs), the key variable:

a.       Table 1 shows that it, has an extreme range (0.008 – 1.01 (or 0.006 – 0.61): order of magnitude difference of over 100!) SO are these really a homogeneous (comparable) set of hospitals? The Cardiology graph in Figure 1 shows this in that specialty, with an outlier over 1 twice about twice as large as any other. Is this the same organisation as the one with the highest PBRs in the two top graphs – where this highest PBR hospital has unusually low quality?

RE: We thank the reviewer for bringing this up. The reason for the extreme range in the PBR/PBR_FTEs is the heterogeneity between the three different specialties. For example, for the variable PBR the range is 0.008-0.486 for oncology, 0.028-1.01 for Cardiology and 0.011- 0.29 for Orthopedics, whereas for PBR_FTEs, the range is 0.006-0.17 for Oncology, 0.014- 0.61 for Cardiology and 0.007-0.23 for orthopedics. Note that the pronounce differences in the PBR ranges between the specialties is the precise reason why we included the interaction terms. For the sake of clarity, we added to Table 1 the differential ranges of the three specialties for both PBR/PBR_FTEs (see P5 and P6 in the revised MS), and in that vein we emphasized this heterogeneity in PBR variation as a justification for using interaction terms in Table 3, i.e. Specialty*PBR or Specialty*PBR_FTEs. (See P9, L332-333).  

Regarding Figure 1; after carefully checking the data which was used to generate figure 1, it appears that while the figures for Cardiology and orthopedics were correctly generated, the syntax that created the figure for Oncology was drawing data from a wrong variables vector. This was corrected and Figure 1 in the revised MS (P11) now includes the correct figure for Oncology. Checking further, we confirm that all regression analyses presented in the paper (Tables 2,3 and 4) used the correct PBR/IHQ data. Note that the revised Figure 1 demonstrates an extreme outlier points (which were marked in yellow) in Cardiology, in Oncology and in Orthopedics. These three outliers indeed represent the same hospital, as the reviewer suspected. In our response to the reviewer’s ii.a comment, we report the results of a robustness check (Cook's distance to estimate the influence of possible outlier data points) which confirms that despite the outliers, quadratic relationships do seem to exist. We also added to Figure 1 a right panel which presents the same graphs for PBR_FTEs for more balanced and complete presentation. Finally, we ran Figure 1 again without the observations that present outliers (see new Figure, S6 in the revised MS) the results reveal a marked quadratic pattern at least in Oncology and Cardiology. We add caveats about the weaknesses of Figure 1 (P11, L361-367).

b.      The Oncology graph in Figure 1 suggests there are some negative PBRs. What does this mean?! 

RE:  We thank the reviewer again for noticing this error. As mentioned in our answer to the previous sub-comments (a), after carefully re-checking the data, it appears that while the figures for Cardiology and orthopedics were correctly generated, the syntax that created the figure for Oncology was drawing data from a wrong variable vector. Notably, we confirmed that all regression analyses have been using the correct data PBR/IHQ. The correct figure for Oncology now appears in the revised MS on P11 instead of the previous one, and as expected, there is no data points that suggest negative PBR.

                   ii.    The fit of the quadratic in PBR:

a.    Although the regression stats (Table 3) show the quadratic term is very statistically, the graphs (Fig 1) make these fits look very dubious. The worst is Cardiology, where the curvilinear relationship appears to be driven entirely by the one outlier (noted in (i). The other two curves also look very sensitive to what look like outliers: bottom right in particular, but maybe also bottom left - (so 1 or 2 hospitals with unusually low-quality metrics). In these top two graphs, are the 95% CIs not consistent with failing to reject a linear fit?

RE: We thank the reviewer for this observation. This comment pertains also to the Oncology graph in Figure 1. As mentioned in our answers to the previous comments (a) and (b), it appears that there was a mistake in the syntax creating the Oncology graph (retrieving data from wrong vectors) while the graphs for Cardiology and orthopedics were correctly generated. The correct graph for Oncology now appears in revised MS in Figure 1 on P11. Given the revised Figure 1, Oncology and Orthopedics seem less vulnerable to outliers, while the graph for Cardiology seems to be potentially driven by the outlier at the right hand of the figure. To test the effect of these outliers (especially in Cardiology, and to a lesser extent in Orthopedics) we employed Cook's distance to estimate the influence of possible outlier data points. We used Stata’s 15.1- Cooksd- postestimation option. Several suspicious observations, for which Cook’s distance was greater than n/4, were identified. Nevertheless, excluding these observations from the regression did not affect the stability of the model and the quadratic relationships between PBR/PBR_FTEs and IHQ remains. Finally, in response to the last part of the reviewer comment’s: “In these top two graphs (i.e. Oncology and Orthopedics), are the 95% CIs not consistent with failing to reject a linear fit?”   the graph representing Oncology seems now more consistent with rejecting a linear fit. It is less clear with respect to Orthopedics. As noted by the reviewer below, this may be a result of the graphs being bivariate, so there are some other variables in play.

We have added these explanations and paid heed to the weaknesses of the graphs on P11, L355-362. The results of the robustness check (Cook's distance) appear on P13, L407-414.  

b.       Are these potentially influential points in the top 2 or all 3 graphs actually the same 2 hospitals?

RE:  As mentioned in our previous responses, the Oncology graph was corrected and does no longer exhibit potentially influential points. The outliers in the graphs of Cardiology and Orthopedics (the one at lower right) represent the same hospital, whereas in Orthopedics the upper right point (high PBR and high qulity) represent a different hospital.

Of course the graphs are bivariate, so there are some others variables in play, and the authors are to be congratulated for showing the graphical visualisation of fits and their confidence intervals. These are important and particularly revealing here rather than just blind tables of regression output - but the authors should have paid heed to the weaknesses the graphs show.

My view is that these issues seriously undermine the credibility of the analysis and so findings.

(i)                      needs consideration and explanation.

RE: As mentioned and explained above, we corrected the graph representing Oncology in Figure 1 (P10). We also added a new part to Figure 1 (the right-hand panel) with parallel graphs which refer to PBR_FTEs (rather than PBR). In this panel the graph for Orthopedics demonstrates a better fit to quadratic relationships and the confidence intervals do not seem consistent with failing to reject a linear fit. We marked in yellow, in Figure 1, the suspicious outliers and discuss the concerns they raise about the fit of the quadratic relationships (P11 L355-361). We also added a supplemental figure (S6) in which Figure 1 is repeated without the organization that is responsible for the outliers.     

(ii) Needs a rethink. The authors should (anyway) consider conventional multiple regression robustness diagnostics for their models. Here, in particular, the influence of individual datapoints could be considered: like the leverage, Cook’s distance …

 RE: As noted above, (in our answer to 2ii), we embraced the reviewer’s advice and employed Cook's distance to estimate the influence of possible outlier data points. We used Stata’s 15.1 Cooksd postestimation option. Several suspicious observations, for which Cook’s distance was greater than n/4, were identified. Nevertheless, excluding these observations from the regression did not affect the stability of the model and the quadratic relationships between PBR/PBR_FTEs and IHQ remains. This serves as a robustness check for the non-linearity pattern estimated (P13, L414-419 in the revised MS). We also note that the relationship still holds when other performance measures are taken into account, namely, survival rate, hospital revenues and net income (P13, L 421-422).

  Minor issues:

The paper is carelessly written – these are hygiene factors!

·         Typos (e.g. “weekly” for weakly, “sub-sect 2.1” should be 2.2, p.5 line 188 j should be i etc)  

RE: These and other typos we identified (for example the header of 3.3) were corrected.

·         Citation style is a mix of Vancouver and Harvard style-families  

RE: Citation style now adheres to Vancouver. We also corrected all references based on the style used by the American Chemical Society as customary in MDPI Journals.

·         Some refs are wrong (e.g. 31)

RE: Reference 31 was corrected to 32 (P2, L50) ref 38 was corrected to 37 (P14, L447).

·         The description of physician intensity is ambiguous in many places: it talks about volumes or numbers of physicians and size of dept I think the ratio is meant

RE:  we now emphasized that it is the ‘physicians per bed’ ratio wherever physician volumes or numbers was mentioned. (e.g. P3, L137; P4, L137; P7, L258, P7, L286; P14,L459, L473, L476, L478)

·         The annual average of publications is ambiguous in the text: average over staff? FTEs?

RE: The annual average of publications was calculated as follow: For each department (i.e. given specialty in a given hospital), we calculated the average numbers of publications per physician (FTE) per year (starting from the year his/her first paper was published). We then employed the median of this averages for the regression analysis. See P5, L 209-211, and the additional explanation in Table 1 P5 for the variable ‘Research volume’.

·         Footnotes should be integrated into the text.

RE: done.

·         Formatting is erratic in places (p. 7 section headings)

RE: This was fixed in the revised MS (P7)

·         The regressions seem to give the sample size as 149, the dataset is said to contain 150 points, but there is no comment about missing data.

RE: the reason for the missing data is that Gundersen Lutheran La Crosse Hospital was not ranked in Cardiology. This is now noted in the text in the revised MS P6, L233-235.
